# Predict models for prolonged ICU stay using APACHE II, APACHE III and SAPS II scores: A Japanese multicenter retrospective cohort study

Daiki Takekawa[1]☯*, Hideki Endo[2]☯, Eiji Hashiba[3]‡, Kazuyoshi Hirota[1]‡

1 Department of Anesthesiology, Graduate School of Medicine, The Hirosaki University, Hirosaki, Japan, 2 Department of Healthcare Quality Assessment, Graduate School of Medicine, The University of Tokyo, Tokyo, Japan, 3 Division of Intensive Care Unit, Hirosaki University Hospital, Hirosaki, Japan

☯ These authors contributed equally to this work.
‡ These authors also contributed equally to this work
* takekawa@hirosaki-u.ac.jp

**Data Availability Statement:** The author's agreement with JIPAD does not allow publishing the data used for this manuscript or sharing it with others. The JIPAD Working Group would cooperate

## Abstract

Prolonged ICU stays are associated with high costs and increased mortality. Thus, early prediction of such stays would help clinicians to plan initial interventions, which could lead to efficient utilization of ICU resources. The aim of this study was to develop models for predicting prolonged stays in Japanese ICUs using APACHE II, APACHE III and SAPS II scores. In this multicenter retrospective cohort study, we analyzed the cases of 85,558 patients registered in the Japanese Intensive care Patient Database between 2015 and 2019. Prolonged ICU stay was defined as an ICU stay of >14 days. Multivariable logistic regression analyses were performed to develop three predictive models for prolonged ICU stay using APACHE II, APACHE III and SAPS II scores, respectively. After exclusions, 79,620 patients were analyzed, 2,364 of whom (2.97%) experienced prolonged ICU stays. Multivariable logistic regression analyses showed that severity scores, BMI, MET/RRT, postresuscitation, readmission, length of stay before ICU admission, and diagnosis at ICU admission were significantly associated with higher risk of prolonged ICU stay in all models. The present study developed predictive models for prolonged ICU stay using severity scores. These models may be helpful for efficient utilization of ICU resources.

## Introduction

Length of stay (LOS) in the intensive care unit (ICU) is an indicator of the efficiency of intensive care, because it is associated with both the costs of intensive care and ICU resource utilization [1,2]. Because ICU resources are limited but the demands have recently increased, we considered that an examination of LOS in the ICU could lead to improvements in the efficiency of intensive care.

in case any fraud or forgery are suspected on manuscripts in which JIPAD data is used. If researchers want to access the data, please contact JIPAD Working Group. E-mail address of JIPAD Working Group is jipad_data@jsicm.org.

**Funding:** The authors received no specific funding for this work.

**Competing interests:** I have read the journal's policy and the authors of this manuscript have the following competing interests: [HE is affiliated with the Department of Healthcare Quality Assessment at the University of Tokyo. The department is a social collaboration department supported by grants from the National Clinical Database, Johnson & Johnson K.K., and Nipro Corporation.] This does not alter our adherence to PLOS ONE policies on sharing data and materials.

Prolonged ICU stay is a serious problem worldwide, because it burdens patients, their families and the economy of their countries with huge expenses, and because it can increase the risk of infection and ICU acquired weakness [3,4]. Moreover, prolonged ICU stay has also been associated with increased mortality and morbidity [5]. Finally, the occupation of ICU beds due to prolonged ICU stay causes delayed admission for other incoming patients [6]. Thus, early prediction of prolonged ICU stay and shortening of LOS in the ICU are needed for cost reduction and the improvement of patient outcomes.

A variety of scoring systems, such as the Acute Physiology and Chronic Health Evaluation (APACHE) score and Simplified Acute Physiology Score (SAPS), have been used to assess disease severity in ICU patients and predict hospital mortalities [7–9]. These are based on the worst data obtained within the first 24 h post-admission and chronic diseases. There are several predictive models for LOS in the ICU, and these models have shown that both SAPS and the APACHE score are significant predictors for LOS in the ICU [10,11]. However, because the medical systems vary widely among countries or regions, further studies from different countries and regions are needed.

In the present retrospective cohort study using the Japanese Intensive care PAtient Database (JIPAD), we sought to develop a model for predicting prolonged ICU stay based on the APACHE II, APACHE III and SAPS II scores. In addition, we evaluated the association between prolonged ICU stay and hospital mortality.

## Materials and methods

### Study procedures and patients

This multi-center, retrospective cohort study was approved by the Ethics Committee of the Hirosaki University Graduate School of Medicine, Hirosaki, Japan, and was publicized on our department homepage (2020–150). Written informed consent from each patient was waived because of the study's retrospective manner, and the Ethics Committee approved the waiver.

We analyzed the data of cases registered in a Japanese ICU database, JIPAD [12], between April 1, 2015 and March 31, 2019. Data were collected at 50 ICUs.

JIPAD was established by the Japanese Society of Intensive Care Medicine in 2014. 94 ICUs participate in JIPAD as of December 31, 2021. Each participating ICU submits data of their ICU patients, such as characteristics, severity scores at ICU admission, therapies in the ICU and outcomes, using on-line data system. Submitted data are routinely monitored and corrected by members of the JIPAD working group to improve the credibility of data. A recent study developed a mortality prediction model for adult patients admitted to ICU in Japan using JIPAD [13]. Further information about JIPAD can be found elsewhere [12].

There were 85,558 patients registered in the JIPAD database over the study period. We excluded patients aged <16 years, burn patients, patients who were admitted to the ICU for single procedures and patients with missing values. Prolonged ICU stay was defined as an ICU stay of >14 days, because special fees for critical care are limited to a period of 14 days in the Japanese health insurance system, even if the status of the patient remains serious for more than 14 days. Additionally, previous studies about prolonged ICU stay that conducted in other countries also used this definition [2,14,15]. Patients with an ICU stay of >14 days were assigned to the prolonged ICU stay group (group P), and those with an ICU stay of ≤14 days were assigned to a non-prolonged ICU stay (group NP).

### Data collection

The following data were obtained from the database: demographic data [gender, age, body mass index (BMI) (category: <18.5, 18.5≤BMI<25, 25≤BMI<35, 35≤), rapid response team/

medical emergency team (RRT/MET), post resuscitation, emergency admission, type of admission (non-operation, elective surgery, or emergency surgery), and readmission], chronic co-morbidity [acquired immunodeficiency syndrome (AIDS), heart failure, respiratory failure, hepatic insufficiency, cirrhosis, acute myeloid leukemia/multiple myeloma, lymphoma, cancer metastasis, hemodialysis, and immunosuppression], LOS before ICU admission, diagnosis at ICU admission, APACHE II, APACHE III and SAPS II scores, acute kidney injury in the first 24 h after ICU admission, mechanical ventilation in the first 24 h after ICU admission, treatment in the ICU [intra-aortic balloon pumping, veno-venous extracorporeal membrane oxygenation, percutaneous cardiopulmonary support, intermittent renal replacement therapy, continuous renal replacement therapy, and plasma exchange], and outcome (LOS in the ICU, LOS in the hospital, ICU death, and hospital death). Definitions of chronic co-morbidity were based on that of APACHE II, APACHE III and/or SAPS II scores.

## Statistical analyses

Patient's characteristic data are presented as the median (25th to 75th percentile) and the number (a percentage of each group). Statistical differences between the study groups were assessed using Chi-squared test for categorical variables and Mann–Whitney U test for continuous variables.

Multivariable logistic regression analyses were performed to develop predictive models for prolonged ICU stay using the APACHE II, APACHE III and SAPS II scores (Models 1–3). Models 1, 2, and 3 included APACHE II, APACHE III and SAPS II scores, respectively. These scores were split into multiple categories: for the APACHE II score, the categories were –10, 11–20, 21–30, 31–40, and 41+; for the APACHE III score, they were –40, 41–80, 81–120, 121–160, and 161+; and for the SAPS II score, they were –25, 25–50, 51–75, 76–100, and 101+. In addition to gender and BMI, readmission, type of admission, LOS before ICU admission, and diagnosis at ICU admission were included to adjust the patient characteristics in these models, because these variables were reported to be associated with prolonged ICU stay [2,14–16]. RRT/MET and post resuscitation were also included, because these were associated with emergency admission or surgery, which is reported to be a risk factor for prolonged ICU stay [2,16]. Several factors that are used for the calculation of these scores or associated with these scores, such as age, type of admission, and chronic disease, were not included. Because there were no patients with post-operative hematological disease in group P, post-operative hematological disease was combined with post-operative metabolic disease in these models. Variance inflation factor (VIF) was used to check for multicollinearity among the variables. VIF >10 indicates the presence of multicollinearity, which requires correction of variable selection. Discrimination was measured using the area under the curve (AUC). The results are expressed as adjusted odds ratios (aORs) with corresponding 95% confidence intervals (CIs).

Moreover, multivariable logistic regression analyses were conducted to evaluate whether prolonged ICU stay is associated with hospital mortality among ICU survivors. The APACHE II, APACHE III and SAPS II scores were included in the respective model to adjust for the disease severity of the patient.

In the present study, we used all available data in the JIPAD database in order to maximize the power and generalizability of the results.

All data analyses were performed with EZR software ver. 1.37 (Saitama Medical Center, Jichi Medical University, Saitama, Japan). P-values<0.05 were considered significant in all tests.

This manuscript adheres to the applicable TRIPOD guideline [17].

## Results

### Characteristics of patients

Of the 85,558 patients, 79,620 patients were finally analyzed after the above-described exclusions (Fig 1). Of the 79,620 patients, 77,256 and 2,364 patients were assigned to groups NP and P, respectively. The prevalence of prolonged ICU stay was 2.97%. The patient characteristics are shown in Tables 1–4. There were significant differences between the two groups in gender (male) (NP group vs. P group; 61.3% vs. 64.7%, p<0.001), BMI (22.4 kg/m$^2$ vs. 22.2 kg/m$^2$, p<0.001), RRT/MET (2.3% vs. 10.3%, p<0.001), post resuscitation (2.3% vs. 9.2%, p<0.001), emergency admission (38.8% vs. 82.8%, p<0.001), readmission (4.1% vs. 12.7%, p<0.001), LOS before ICU admission (but there was no significant difference in the median LOS before ICU admission), type of admission, outcome (Table 1), diagnosis, chronic disease except AIDS (Table 2), therapy in the ICU (Table 3), APACHE II score (13 vs. 23, p<0.001), APACHE III score (51 vs. 86, p<0.001), and SAPS II score (26 vs. 50, p<0.001) (Table 4).

### Distribution of patients stratified by length of stay in the ICU

The mean ICU-LOS of all patients was 3.2 (standard deviation: ±5.9) days, with a median of 1 day (25th to 75th percentile: 1–3). 2,749 (3.5%), 41,201(51.7%), 28,440 (35.7%), 4,866 (6.1%), 1255 (1.6%), and 1,111 (1.4%) patients spent 0, 1, 2–7, 8–14, 15–21, and 22+ days in the ICU, respectively (Fig 2).

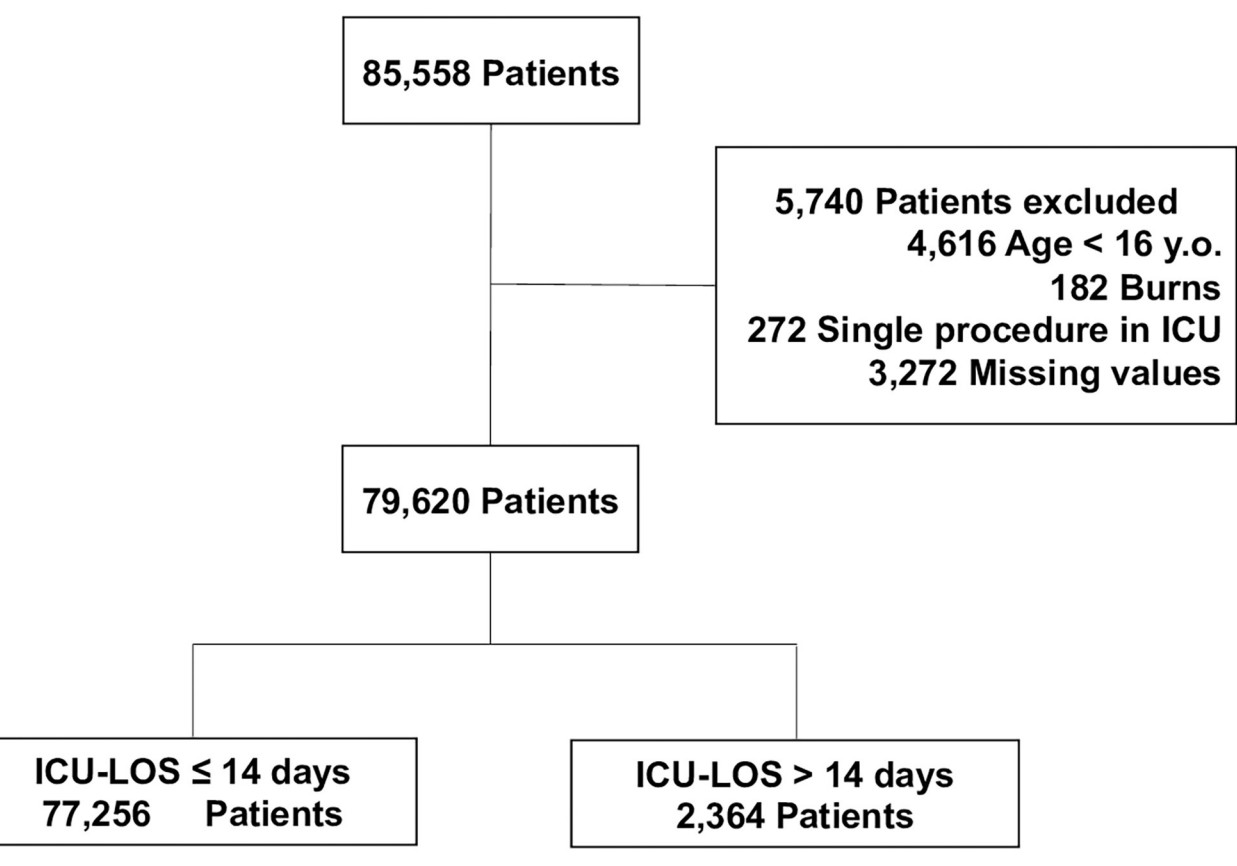

**Fig 1. Flowchart outlining patient selection and grouping process.**

**Table 1. Patient characteristics.**

| | Group NP | Group P | P value |
|---|---|---|---|
| N | 77,256 | 2,364 | |
| Male | 47,357 (61.3%) | 1,530 (64.7%) | <0.001* |
| Age (year) | 70 (60, 78) | 70 (58, 77) | 0.155 |
| BMI (kg/m$^2$) | 22.4 (20.0, 25.0) | 22.2 (19.5, 25.0) | <0.001* |
| BMI (kg/m$^2$) | | | <0.001* |
| <18.5 | 10,440 (13.5%) | 422 (17.9%) | |
| 18.5≤, < 25 | 47,274 (61.2%) | 1,367 (57.8%) | |
| 25 ≤, < 35 | 18,572 (24.0%) | 539 (22.8%) | |
| 35 ≤ | 970 (1.3%) | 36 (1.5%) | |
| RRT/MET | 1,756 (2.3%) | 243 (10.3%) | <0.001* |
| Post resuscitation | 1,815 (2.3%) | 217 (9.2%) | <0.001* |
| Emergency Admission | 29,944 (38.8%) | 1,958 (82.8%) | <0.001* |
| Type of admission | | | <0.001* |
| Non-operation | 19,579 (25.3%) | 1,455 (61.5%) | |
| Elective surgery | 47,884 (62.0%) | 385 (16.3%) | |
| Emergency surgery | 9,793 (12.7%) | 524 (22.2%) | |
| Readmission | 3197 (4.1%) | 301 (12.7%) | <0.001* |
| LOS before ICU admission(days) | 2 (1, 6) | 2 (0, 13) | 0.861 |
| LOS before ICU admission | | | <0.001* |
| 0 | 17,326 (22.4%) | 880 (37.2%) | |
| 1–7 | 45,953 (59.5%) | 728 (30.8%) | |
| 8–14 | 5,985 (7.7%) | 212 (9.0%) | |
| 15+ | 7,992 (10.3%) | 544 (23.0%) | |
| AKI in first 24h | 2044 (2.6%) | 282 (11.9%) | <0.001* |
| MV in first 24h | 26,223 (33.9%) | 1,920 (81.2%) | <0.001* |
| Outcome | | | |
| ICU-LOS (days) | 1 (1, 3) | 21 (17, 29) | <0.001* |
| Hospital-LOS (days) | 20 (11, 36) | 66 (41, 110) | <0.001* |
| ICU-Death | 2,513 (3.3%) | 444 (1.8%) | <0.001* |
| Hospital-Death | 5,601 (7.2%) | 926 (39.2%) | <0.001* |

Differences between the group NP and group P were estimated using chi-squared test for categorical variables and Mann–Whitney U test for continuous variables. Data are presented as number (percentage of each group) or median (25th to 75th percentile). BMI: Body mass index, RRT/MET: Rapid response team/medical emergency team, LOS: Length of stay, AKI: Acute kidney injury, MV: Mechanical ventilation,
*: Statistical significance.

## Severity scores and prolonged ICU stay

APACHE II, APACHE III and SAPS II scores were significantly higher in group P than in group NP (Table 4).

The results of multivariable logistic regression analyses to develop predictive models for prolonged ICU stay using the APACHE II, APACHE III and SAPS II scores are shown in Table 5. ROC curves for each predictive model are shown in Fig 4. When the type of admission was included in these models, the VIF values of it and diagnosis were more than 10 in all models. Thus, the type of admission was not included. In these models, there were no VIF values of 10 or higher, indicating that there was no collinearity.

**Table 2. Diagnosis and chronic disease.**

| Diagnosis | | | <0.001* |
|---|---|---|---|
| •Post-operative | | | |
| Cardiovascular disease | 16,148 (20.9%) | 459 (19.4%) | |
| Respiratory disease | 8,636 (11.2%) | 32 (1.4%) | |
| Digestive disease | 14,941 (19.3%) | 209 (8.8%) | |
| Neurological disease | 9,020 (11.7%) | 140 (5.9%) | |
| Trauma | 721 (0.9%) | 36 (1.5%) | |
| Metabolic disease | 549 (0.7%) | 4 (0.2%) | |
| Hematological disease | 24 (0.0%) | 0 (0.0%) | |
| Urinary disease | 2,780 (3.6%) | 8 (0.3%) | |
| Muscle/bone/skin disease | 3,139 (4.1%) | 23 (1.0%) | |
| Obstetrics/gynecological disease | 1,728 (2.2%) | 5 (0.2%) | |
| •Non-operative | | | |
| Cardiovascular disease | 7,770 (10.1%) | 380 (16.1%) | |
| Respiratory disease | 3,478 (4.5%) | 521 (22.0%) | |
| Digestive disease | 1,875 (2.4%) | 139 (5.9%) | |
| Neurological disease | 2,145 (2.8%) | 139 (5.9%) | |
| Sepsis | 1,277 (1.7%) | 108 (4.6%) | |
| Trauma | 811 (1.0%) | 49 (2.1%) | |
| Metabolic disease | 1,119 (1.4%) | 22 (0.9%) | |
| Hematological disease | 220 (0.3%) | 20 (0.8%) | |
| Urinary disease | 349 (0.5%) | 15 (0.6%) | |
| Muscle/bone/skin disease | 251 (0.3%) | 34 (1.4%) | |
| Others | 275 (0.4%) | 21 (0.9%) | |
| Chronic diseases | | | |
| AIDS | 35 (0.0%) | 2 (0.1%) | 0.301 |
| Heart Failure | 1,004 (1.3%) | 95 (4.0%) | <0.001* |
| Respiratory Failure | 880 (1.1%) | 88 (3.7%) | <0.001* |
| Hepatic Insufficiency | 322 (0.4%) | 41 (1.7%) | <0.001* |
| Cirrhosis | 988 (1.3%) | 58 (2.5%) | <0.001* |
| AML/MM | 429 (0.6%) | 59(2.5%) | <0.001* |
| Lymphoma | 494 (0.6%) | 2 (2.2%) | <0.001* |
| Cancer metastasis | 3,334 (4.3%) | 72 (3.0%) | 0.002* |
| HD | 3,842 (5.0%) | 204 (8.6%) | <0.001* |
| Immunosuppression | 4,266 (5.5%) | 304 (12.9%) | <0.001* |

Differences between the group NP and group P were estimated using chi-squared test. Data are presented as number (percentage of each group). AIDS: Acquired immunodeficiency syndrome, AML/MM: Acute myeloid leukemia/multiple myeloma, HD: Hemodialysis,

*: Statistical significance.

Model 1 showed that the APACHE II score (categories: 11–20, 21–30, 31–40, and 41+), BMI (category: 25–35 kg/m$^2$), MET/RRT, post resuscitation, readmission, LOS before ICU admission (category: 15+ days), and diagnosis (categories: non-operative respiratory disease, non-operative neurological disease, non-operative trauma, and non-operative muscle/bone/skin disease) were significantly associated with higher risk of prolonged ICU stay (Table 5, Model 1). LOS before ICU admission (category: 1–7 days) and diagnosis (categories: post-operative respiratory disease, post-operative digestive disease, post-operative neurological

**Table 3. Therapy in ICU.**

| | | | |
|---|---|---|---|
| IABP | 1,325 (1.7%) | 274 (11.6%) | <0.001* |
| V-V ECMO | 67 (0.1%) | 76 (3.2%) | <0.001* |
| PCPS | 438 (0.6%) | 213 (9.0%) | <0.001* |
| Tracheostomy | 927 (1.2%) | 1,003 (42.4%) | <0.001* |
| IRRT | 2,848 (3.7%) | 487 (20.6%) | <0.001* |
| CRRT | 3,254 (4.2%) | 951 (40.2%) | <0.001* |
| PE | 235 (0.3%) | 104 (4.4%) | <0.001* |

Differences between the group NP and group P were estimated using chi-squared test. Data are presented as number (percentage of each group). IABP: Intra-aortic balloon pumping, V-V ECMO: Veno-venous extracorporeal membrane oxygenation, PCPS: Percutaneous cardiopulmonary support, IRRT: Intermittent renal replacement therapy, CRRT: Continuous renal replacement therapy, PE: Plasma exchange,

*: Statistical significance.

disease, post-operative urinary disease, post-operative muscle/bone/skin disease, post-operative obstetrics/gynecological disease, and non-operative metabolic disease) were significantly associated with lower risk of prolonged ICU stay (Table 5, Model 1). The AUC value of this model was 0.827 (Fig 3, Model 1).

**Table 4. Severity scores.**

| | | | |
|---|---|---|---|
| APACHE II score | 13 (10, 18) | 23 (18, 29) | <0.001* |
| APACHE II score | | | <0.001* |
| −10 | 20,340 (26.3%) | 83 (3.5%) | |
| 11−20 | 44,371 (57.4%) | 827 (35.0%) | |
| 21−30 | 9,387 (12.2%) | 967 (40.9%) | |
| 31−40 | 2,275 (2.9%) | 404 (17.1%) | |
| 41+ | 883 (1.1%) | 83 (3.5%) | |
| APACHE III score | 51 (39, 68) | 86 (68, 109) | <0.001* |
| APACHE III score | | | <0.001* |
| −40 | 22,337 (28.9%) | 61 (2.6%) | |
| 41−80 | 43,943 (56.9%) | 946 (40.0%) | |
| 81−120 | 8,295 (10.7%) | 943 (39.9%) | |
| 121−160 | 1,931 (2.5%) | 353 (14.9%) | |
| 161+ | 750 (1.0%) | 61 (2.6%) | |
| SAPS II score | 26 (19, 37) | 50 (39, 64) | <0.001* |
| SAPS II score | | | <0.001* |
| −25 | 36,226 (46.9%) | 111 (4.7%) | |
| 26−50 | 32,624 (42.2%) | 1,102 (46.6%) | |
| 51−75 | 6,308 (8.2%) | 878 (37.1%) | |
| 76−100 | 1,713 (2.2%) | 254 (10.7%) | |
| 101+ | 385 (0.5%) | 19 (0.8%) | |

Differences between the group NP and group P were estimated using chi-squared test for categorical variables and Mann–Whitney U test for continuous variables. Data are presented as number (percentage of each group) or median (25th to 75th percentile). APACHE: Acute physiology and chronic health evaluation, SAPS: Simplified acute physiology score,

*: Statistical significance.

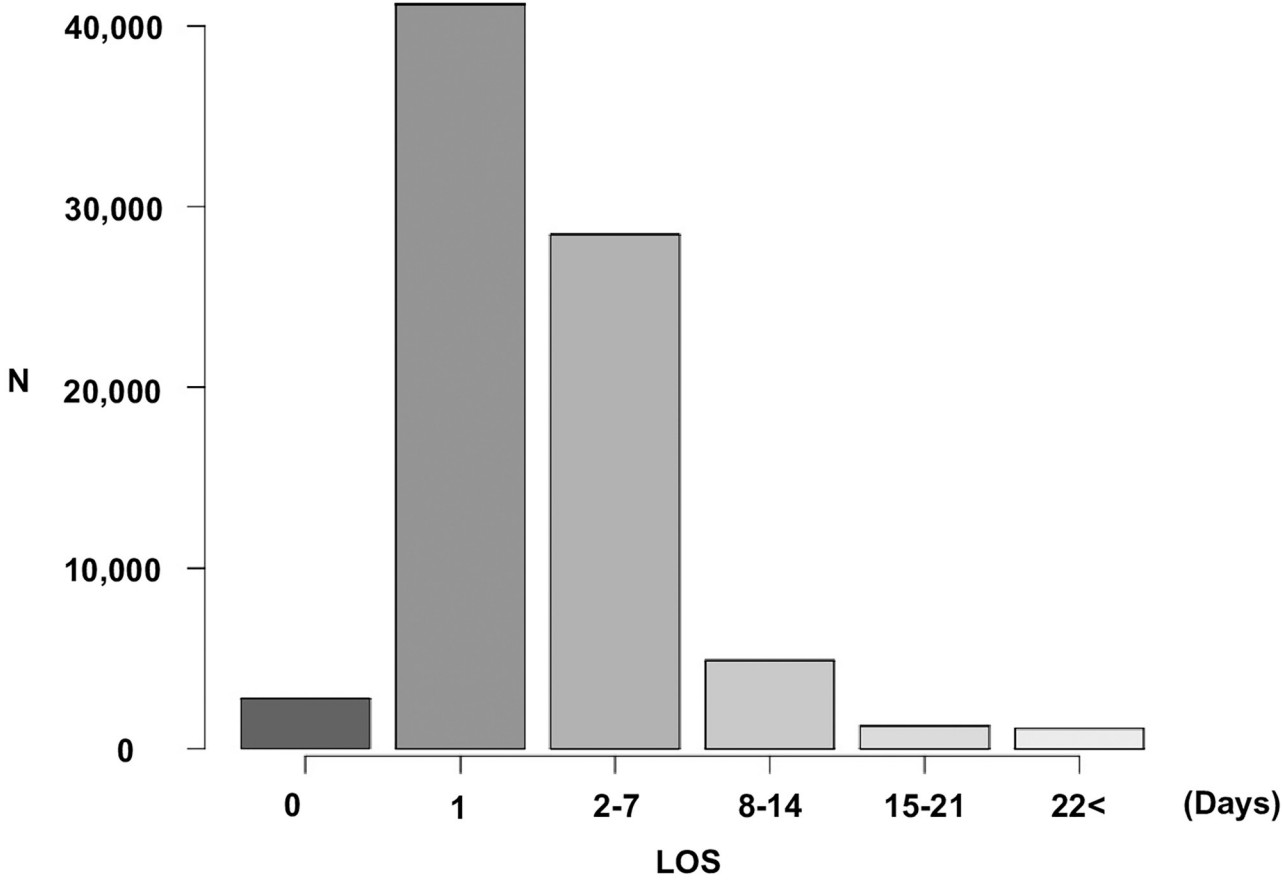

**Fig 2. Distribution of patients stratified by length of stay in ICU.**

Model 2 showed that the APACHE III score (categories: 41–80, 81–120, 121–160, 160+), BMI (category: 25–35 kg/m$^2$), MET/RRT, post resuscitation, readmission, LOS before ICU admission (category: 15+ days), and diagnosis (categories: post-operative trauma, non-operative respiratory disease, non-operative neurological disease, non-operative trauma, and non-operative muscle/bone/skin disease) were significantly associated with higher risk of prolonged ICU stay (Table 5, Model 2). LOS before ICU admission (category: 1–7 days) and diagnosis (categories: post-operative respiratory disease, post-operative digestive disease, post-operative urinary disease, post-operative muscle/bone/skin disease, post-operative obstetrics/gynecological disease and non-operative metabolic disease) were significantly associated with lower risk of prolonged ICU stay (Table 5, Model 2). The AUC value of this model was 0.833 (Fig 3, Model 2).

Model 3 showed that the SAPS II score (categories: 26–50, 51–75, 75–100, 101+), BMI (category: 25–35 kg/m$^2$), MET/RRT, post resuscitation, readmission, LOS before ICU admission (category: 15+ days), and diagnosis (categories: non-operative respiratory disease, non-operative neurological disease, non-operative trauma, and non-operative muscle/bone/skin disease) were significantly associated with higher risk of prolonged ICU stay (Table 5, Model 3). LOS before ICU admission (category: 1–7 days) and diagnosis (categories: post-operative respiratory disease, post-operative digestive disease, post-operative urinary disease, post-operative muscle/bone/skin disease, post-operative obstetrics/gynecological disease, and non-operative

**Table 5. Predictive model for prolonged ICU stay using APCHE II, APCHE III and SAPS II score.**

| Model 1 | | | |
|---|---|---|---|
| | aOR | 95% CI | P value |
| (Intercept) | 0.007 | 0.005, 0.009 | <0.001* |
| APACHE II | | | |
| ~10 | reference | | |
| 11–20 | 3.680 | 2.930, 4.620 | <0.001* |
| 21–30 | 12.60 | 10.00, 16.00 | <0.001* |
| 31–40 | 17.90 | 13.90, 23.20 | <0.001* |
| 41+ | 9.100 | 6.530, 12.70 | <0.001* |
| Male | 1.060 | 0.973, 1.160 | 0.172 |
| BMI (kg/m$^2$) | | | |
| $18.5 \leq, < 25$ | reference | | |
| $< 18.5$ | 0.959 | 0.853, 1.080 | 0.479 |
| $25 \leq, < 35$ | 1.140 | 1.030, 1.270 | 0.014* |
| $35 \leq$ | 1.340 | 0.942, 1.900 | 0.103 |
| RRT/MET | 1.240 | 1.060, 1.470 | <0.001* |
| Post resuscitation | 1.330 | 1.090, 1.610 | <0.001* |
| Readmisson | 1.290 | 1.100, 1.500 | <0.001* |
| LOS before ICU admission (days) | | | |
| 0 | reference | | |
| 1–7 | 0.756 | 0.673, 0.849 | <0.001* |
| 8–14 | 0.969 | 0.820, 1.150 | 0.714 |
| 15+ | 1.280 | 1.120, 1.470 | <0.001* |
| Diagnosis | | | |
| •Post-operative | | | |
| Cardiovascular disease | reference | | |
| Respiratory disease | 0.209 | 0.146, 0.300 | <0.001* |
| Digestive disease | 0.550 | 0.466, 0.651 | <0.001* |
| Neurological disease | 0.682 | 0.561, 0.829 | <0.001* |
| Trauma | 1.260 | 0.880, 1.810 | 0.201 |
| Metabolic/hematological disease | 0.431 | 0.160, 1.160 | 0.096 |
| Urinary disease | 0.145 | 0.072, 0.292 | <0.001* |
| Muscle/bone/skin disease | 0.285 | 0.187, 0.435 | <0.001* |
| Obstetrics/gynecological disease | 0.185 | 0.076, 0.448 | <0.001* |
| •Non-operative | | | |
| Cardiovascular disease | 0.895 | 0.755, 1.06 | 0.201 |
| Respiratory disease | 2.070 | 1.780, 2.400 | <0.001* |
| Digestive disease | 1.170 | 0.947, 1.440 | 0.180 |
| Neurological disease | 1.280 | 1.040, 1.580 | 0.020* |
| Sepsis | 1.020 | 0.810, 1.290 | 0.854 |
| Trauma | 1.490 | 1.080, 2.040 | 0.014* |
| Metabolic disease | 0.335 | 0.216, 0.52 | <0.001* |
| Hematological disease | 1.130 | 0.700, 1.83 | 0.615 |
| Urinary disease | 0.783 | 0.459, 1.340 | 0.369 |
| Muscle/bone/skin disease | 2.030 | 1.380, 2.990 | <0.001* |
| Others | 1.260 | 0.793, 2.020 | 0.324 |
| Model 2 | | | |
| | aOR | 95% CI | P value |

(*Continued*)

**Table 5.** (Continued)

| | aOR | 95% CI | P value |
|---|---|---|---|
| (Intercept) | 0.004 | 0.003, 0.006 | <0.001* |
| APACHE III score | | | |
| –40 | reference | | |
| 41–80 | 6.280 | 4.820, 8.170 | <0.001* |
| 81–120 | 21.20 | 16.20, 27.80 | <0.001* |
| 121–160 | 28.50 | 21.30, 38.20 | <0.001* |
| 161+ | 12.50 | 8.540, 18.30 | <0.001* |
| Male | 1.040 | 0.948, 1.130 | 0.426 |
| BMI (kg/m$^2$) | | | |
| 18.5≤, < 25 | reference | | |
| <18.5 | 0.955 | 0.850, 1.070 | 0.445 |
| 25 ≤, < 35 | 1.160 | 1.040, 1.290 | 0.006* |
| 35 ≤ | 1.380 | 0.969, 1.960 | 0.074 |
| RRT/MET | 1.250 | 1.060, 1.440 | 0.007* |
| Post resuscitation | 1.250 | 1.020, 1.520 | 0.030* |
| Readmisson | 1.240 | 1.060, 1.440 | 0.007* |
| LOS before ICU admission (days) | | | |
| 0 | reference | | |
| 1–7 | 0.773 | 0.688, 0.869 | <0.001* |
| 8–14 | 0.955 | 0.807, 1.130 | 0.588 |
| 15+ | 1.290 | 1.130, 1.480 | <0.001* |
| Diagnosis | | | |
| •Post-operative | | | |
| Cardiovascular disease | reference | | |
| Respiratory disease | 0.231 | 0.161, 0.331 | <0.001* |
| Digestive disease | 0.513 | 0.434, 0.607 | <0.001* |
| Neurological disease | 0.850 | 0.698, 1.030 | 0.105 |
| Trauma | 1.470 | 1.020, 2.100 | 0.003* |
| Metabolic/hematological disease | 0.514 | 0.190, 1.390 | 0.190 |
| Urinary disease | 0.156 | 0.077, 0.315 | <0.001* |
| Muscle/bone/skin disease | 0.287 | 0.188, 0.438 | <0.001* |
| Obstetrics/gynecological disease | 0.198 | 0.082, 0.480 | <0.001* |
| •Non-operative | | | |
| Cardiovascular disease | 0.969 | 0.818, 1.150 | 0.720 |
| Respiratory disease | 2.260 | 1.950, 2.620 | <0.001* |
| Digestive disease | 1.200 | 0.970, 1.470 | 0.943 |
| Neurological disease | 1.430 | 1.160, 1.760 | <0.001* |
| Sepsis | 1.060 | 0.837, 1.330 | 0.648 |
| Trauma | 1.710 | 1.240, 2.350 | <0.001* |
| Metabolic disease | 0.362 | 0.233, 0.562 | <0.001* |
| Hematological disease | 1.210 | 0.751, 1.960 | 0.429 |
| Urinary disease | 0.819 | 0.479, 1.400 | 0.465 |
| Muscle/bone/skin disease | 2.080 | 1.420, 3.060 | <0.001* |
| Others | 1.450 | 0.909, 2.310 | 0.119 |
| Model 3 | | | |
| | aOR | 95% CI | P value |
| (Intercept) | 0.004 | 0.003, 0.005 | <0.001* |
| SAPS II score | | | |

(*Continued*)

**Table 5.** (Continued)

| | aOR | CI | p |
|---|---|---|---|
| –25 | reference | | |
| 26–50 | 7.600 | 6.210, 9.310 | <0.001* |
| 51–75 | 22.60 | 18.30, 28.00 | <0.001* |
| 76–100 | 21.30 | 16.60, 27.30 | <0.001* |
| 101+ | 6.830 | 4.090, 11.40 | <0.001* |
| Male | 1.050 | 0.960, 1.150 | 0.285 |
| BMI (kg/m$^2$) | | | |
| 18.5 ≤, < 25 | reference | | |
| < 18.5 | 0.967 | 0.860, 1.090 | 0.577 |
| 25 ≤, < 35 | 1.150 | 1,030, 1.270 | 0.001* |
| 35 ≤ | 1.380 | 0.970, 1.970 | 0.073 |
| RRT/MET | 1.220 | 1.030, 1.440 | 0.019* |
| Post resuscitation | 1.360 | 1.120, 1.660 | 0.002* |
| Readmisson | 1.200 | 1.030, 1.400 | 0.020* |
| LOS before ICU admission (days) | | | |
| 0 | reference | | |
| 1–7 | 0.904 | 0.806, 1.01 | 0.084 |
| 8–14 | 1.12 | 0.944, 1.320 | 0.198 |
| 15+ | 1.520 | 1.320, 1.740 | <0.001* |
| Diagnosis | | | |
| •Post-operative | | | |
| Cardiovascular disease | reference | | |
| Respiratory disease | 0.297 | 0.207, 0.427 | <0.001* |
| Digestive disease | 0.575 | 0.486, 0.680 | <0.001* |
| Neurological disease | 0.866 | 0.710, 1.060 | 0.156 |
| Trauma | 1.270 | 0.884, 1.810 | 0.199 |
| Metabolic/hematological disease | 0.596 | 0.220, 1.610 | 0.308 |
| Urinary disease | 0.222 | 0.110, 0.449 | <0.001* |
| Muscle/bone/skin disease | 0.354 | 0.232, 0.541 | <0.001* |
| Obstetrics/gynecological disease | 0.275 | 0.113, 0.669 | <0.001* |
| •Non-operative | | | |
| Cardiovascular disease | 0.915 | 0.774, 1.080 | 0.300 |
| Respiratory disease | 2.320 | 2.000, 2.690 | <0.001* |
| Digestive disease | 1.290 | 1.050, 1.759 | 0.015* |
| Neurological disease | 1.340 | 1.090, 1.650 | 0.006* |
| Sepsis | 1.150 | 0.915, 1.450 | 0.228 |
| Trauma | 1.620 | 1.180, 2.220 | 0.003* |
| Metabolic disease | 0.386 | 0.248, 0.599 | <0.001* |
| Hematological disease | 1.300 | 0.803, 2.100 | 0.286 |
| Urinary disease | 0.949 | 0.555, 1.620 | 0.847 |
| Muscle/bone/skin disease | 2.410 | 1.640, 3.550 | <0.001* |
| Others | 1.540 | 0.969, 2.460 | 0.068 |

Multivariate logistic regression analyses were performed to develop predictive models for prolonged ICU stay using APCHE II, APCHE III and SAPS II score (Model 1–3). Model 1, 2, and 3 included APCHE II, APCHE III and SAPS II score, respectively. No variance inflation factor value was up to 10, indicating that there was no collinearity in the model. aOR: Adjusted odds ratio, CI: Confidence interval, APACHE: Acute physiology and chronic health evaluation, SAPS: Simplified acute physiology score, BMI: Body mass index, RRT/MET: Rapid response team/ medical emergency team, LOS: Length of stay,

*: Statistical significance.

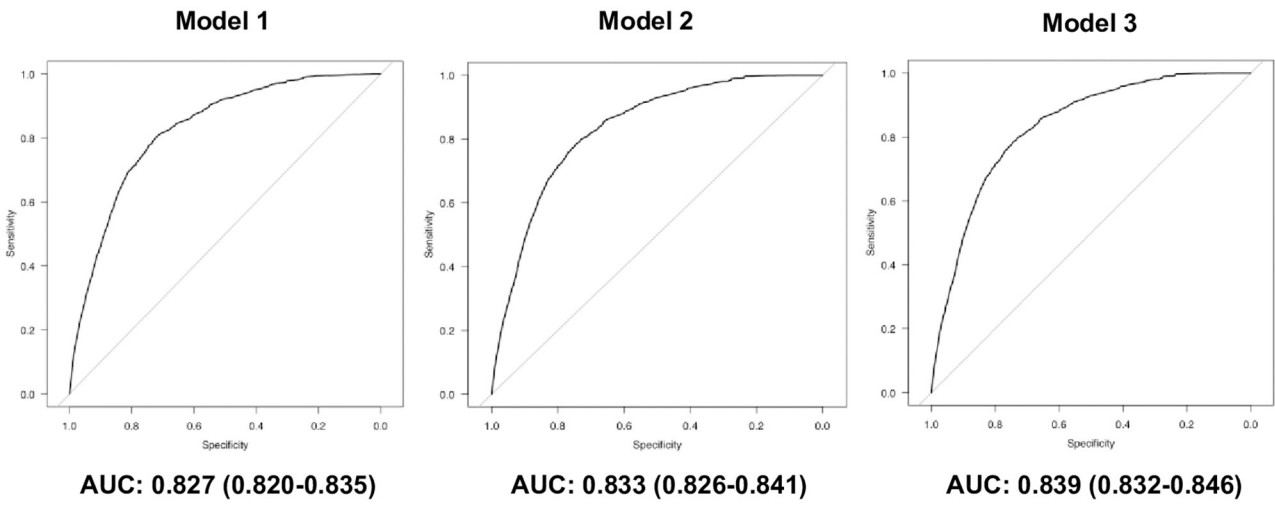

**Fig 3. ROC curves for each model to predict prolonged ICU stay.**

metabolic disease) were significantly associated with lower risk of prolonged ICU stay (Table 5, Model 3). The AUC value of this model was 0.839 (Fig 3, Model 3).

### Prolonged ICU stay and hospital mortality

Hospital mortality of group P was significantly higher than that of group NP (7.2% vs. 39.2%, p<0.001) (Table 1). Multivariable logistic regression analyses showed that prolonged ICU stay was significantly associated with an increased hospital mortality after adjusting for the severity scores (Table 6, Models 1–3). ROC curves for each model are shown in Fig 4. The aORs for hospital mortality increased as the LOS in the ICU increased except 0 day LOS.

### Discussion

The present study developed a predictive model for prolonged ICU stay using APACHE II, APACHE III and SAPS II scores. In addition to each severity score, BMI (category: 25–35 kg/m$^2$), MET/RRT, post resuscitation, readmission, LOS before ICU admission (category: 15 + days), and diagnosis (categories: non-operative respiratory disease, non-operative neurological disease, non-operative trauma, and non-operative muscle/bone/skin disease) were significantly associated with higher risk of prolonged ICU stay in all models. The AUC values for all logistic regression models were more than 0.8, which means that the discrimination abilities of these models were good. Moreover, prolonged ICU stay was significantly associated with an increased hospital mortality.

The present study showed that the prevalence of prolonged ICU stay was 2.97% in Japan. On the other hand, previous studies showed that the prevalence was 4–11% [2,14,15], even though these studies defined a prolonged ICU stay as >14 days just as in the present study. The prevalence of prolonged ICU stay in the present study was the lowest of these four studies. This might be due to the differences in patients' severity and type of admission among the studies. Whereas the mean (±standard deviation) APACHE II score was 15.2±7.4 in the present study, the APACHE II scores in the studies by Arabi et al., Laupland et al., and Zampieri et al. were 19±9, 24.9±8.8, and 22.60±5.21, respectively [2,14,15]. Moreover, the proportion of admission to the ICU after elective surgery was 60.4% in the present study, which was higher

**Table 6. Multivariate logistic regression model to predict hospital-death after adjusting severity score.**

Model 1

|  | aOR | 95% CI | P value |
| --- | --- | --- | --- |
| (Intercept) | 0.002 | 0.002, 0.003 | <0.001* |
| ICU-LOS |  |  |  |
| 1 (day) | reference |  |  |
| 0 (day) | 2.350 | 1.880, 2.950 | <0.001* |
| 2–7 (days) | 1.530 | 1.390, 1.690 | <0.001* |
| 8–14 (days) | 2.340 | 2.060, 2.660 | <0.001* |
| 15–21 (days) | 3.310 | 2.750, 3.990 | <0.001* |
| 22- (days) | 4.230 | 3.480, 3.990 | <0.001* |
| APACHE II score | 1.160 | 1.150, 1.160 | <0.001* |

Model 2

|  | aOR | 95% CI | P value |
| --- | --- | --- | --- |
| (Intercept) | 0.002 | 0.002, 0.002 | <0.001* |
| ICU-LOS |  |  |  |
| 1 (day) | reference |  |  |
| 0 (day) | 2.380 | 1.900, 2.990 | <0.001* |
| 2–7 (days) | 1.430 | 1.300, 1.580 | <0.001* |
| 8–14 (days) | 2.090 | 1.840, 2.380 | <0.001* |
| 15–21 (days) | 2.930 | 2.430, 3.540 | <0.001* |
| 22- (days) | 3.670 | 3.010, 4.460 | <0.001* |
| APACHE III score | 1.040 | 1.040, 1.040 | <0.001* |

Model 3

|  | aOR | 95% CI | P value |
| --- | --- | --- | --- |
| (Intercept) | 0.003 | 0.003, 0.004 | <0.001* |
| ICU-LOS |  |  |  |
| 1 (day) | reference |  |  |
| 0 (day) | 1.900 | 1.520, 2.380 | <0.001* |
| 2–7 (days) | 1.340 | 1.220, 1.840 | <0.001* |
| 8–14 (days) | 1.890 | 1.660, 2.160 | <0.001* |
| 15–21 (days) | 2.790 | 2.320, 3.360 | <0.001* |
| 22- (days) | 3.610 | 2.980, 4.380 | <0.001* |
| APACHE III score | 1.070 | 1.060, 1.070 | <0.001* |

Multivariate logistic regression analyses were conducted to evaluate whether prolonged ICU stay is associated with hospital mortality among ICU survivors. APCHE II, APCHE III and SAPS II score were included to each model to adjust patient's severity. No variance inflation factor value was up to 10, indicating that there was no collinearity in the model. OR: Odds ratio, CI: Confidence interval, APACHE: Acute physiology and chronic health evaluation, SAPS: Simplified acute physiology score, LOS: Length of stay,

*: Statistical significance.

than those in the other studies. As mentioned in the materials and methods section, in the Japanese health insurance system, special fees for critical care are limited to a period of 14 days, even if the status of the patient remains serious for more than 14 days. Thus, the rules of the Japanese health insurance system might affect the prevalence of prolonged ICU stay in Japan.

To the best of our knowledge, this is the first study to develop models for predicting prolonged ICU stays in Japan. We adjusted the disease severity of patient's using the APACHE and SAPS scores. These scores have been reported to be significant predictors for LOS in the

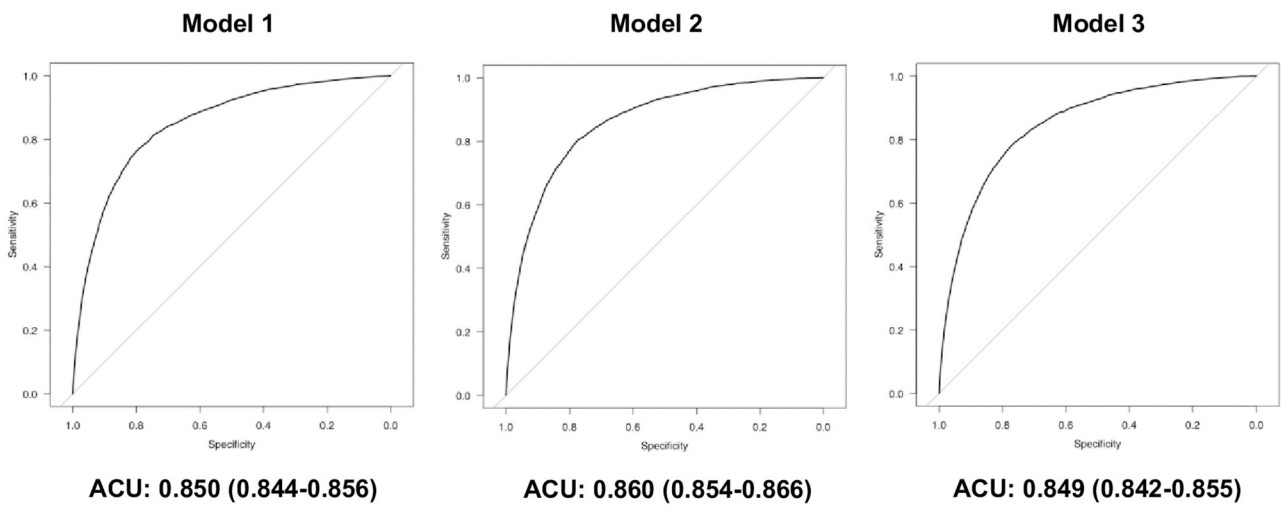

**Fig 4. ROC curves for each model to predict hospital death.**

ICU [9,10,16,18,19]. For purposes of the present analysis, we divided each severity score into five categories. Four of the categories were significantly associated with a higher risk of prolonged ICU stay compared to the lowest category. Although the aORs increased as the severity score category increased up to the 4th category, the aORs decreased in the highest (5th) category compared to the 4th category. This was due to the fact that patients with extremely severe disease are likely to die early in their ICU stay, and thus their LOS in the ICU tends to be short [20,21]. Similarly, BMI of 25–35 kg/m$^2$ was associated with a higher risk of prolonged ICU stay, but BMI of 35+ kg/m$^2$ was not. Readmisson and an LOS before ICU admission of 15 + days were significantly associated with higher risk of prolonged ICU stay in the present study, which is consistent with previous studies [2,15,20]. Although our PubMed search did not uncover any studies that described the relationship between prolonged ICU stay and either RRT/MET or post resuscitation, two studies showed that emergency admission and surgery were significantly associated with higher risk of prolonged ICU stay [2,20]. We did not include them in our models, because they are used for calculation of severity scores or are associated with these scores. Regarding diagnosis at ICU admission, previous studies reported that non-operative respiratory disease, non-operative/post-operative trauma and non-operative neurological disease were significantly associated with higher risk of prolonged ICU stay [2,16]. In addition, the present study showed that non-operative muscle/bone/skin disease were significantly associated with higher risk of prolonged ICU stay. These models may be useful for helping clinicians to identify patients who are likely to have a prolonged ICU stay, and thereby for helping clinicians to control the allocation of ICU beds for effective resource utilization. In addition, this result may be used to calculate claimable special fees for critical care after 14 days in Japan to better reflect the disease severity of the patient compared to the present health insurance system.

This study also showed that prolonged ICU stay was significantly associated with increased hospital mortality. This result was consistent with a previous retrospective cohort study which reported that 1–year mortality increased with increasing LOS in the ICU, and a clear LOS cut-off at which mortality rates significantly change was not found [5]. On the other hand, a retrospective cohort study demonstrated that LOS in the ICU was not significantly associated with a higher risk of hospital mortality [22]. This was likely to be due to the difference in definition

of prolonged ICU stay. In the study by Williams et al., a prolonged ICU stay was defined as >10 days [22]. Another study reported that mortality increased with LOS in the ICU up to 10 days, but remained stable thereafter [23]. Thus, because the association between prolonged ICU stay and mortality remains controversial, multi-center prospective cohort studies are warranted to address this topic.

The present study has several limitations. First, as this was a retrospective observational study, there might be undetected confounding factors that affected the results. Second, as we used a Japanese national database, possible ethnic and medical system differences should be considered. Third, as patients were not followed after discharge from the hospital, we could not evaluate 1–year mortality or long-term outcomes.

## Conclusions

The present study developed predictive models for prolonged ICU stay using the APACHE II, APACHE III and SAPS II scores. In addition, prolonged ICU stay was significantly associated with an increased hospital mortality. These findings may be helpful for the efficient utilization of ICU resources.

## Acknowledgments

We thank for all of the institutions participating JIPAD and the JIPAD working group for their contribution.

## Author Contributions

**Conceptualization:** Daiki Takekawa.

**Formal analysis:** Daiki Takekawa, Hideki Endo.

**Supervision:** Hideki Endo, Eiji Hashiba, Kazuyoshi Hirota.

**Writing – original draft:** Daiki Takekawa.

**Writing – review & editing:** Hideki Endo, Eiji Hashiba, Kazuyoshi Hirota.

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
