## [Decision Letter · Decision Letter 0]

30 Mar 2022

PONE-D-22-01670Predict models for prolonged ICU stay using APACHE II, APACHE III

and SAPS II scores: A Japanese multicenter retrospective cohort studyPLOS ONE

Dear Dr. Takekawa,

Thank you for submitting your manuscript to PLOS ONE. After careful consideration, we feel that it has merit but does not fully meet PLOS ONE’s publication criteria as it currently stands. Therefore, we invite you to submit a revised version of the manuscript that addresses the points raised during the review process.

Reviewer #2 was very critical in evaluating your work, while reviewer #1 gave some helpful comments. If you choose to resubmit your work, you need to fully addressed the concerns raised by the reviewers.

We look forward to receiving your revised manuscript.

Kind regards,

Yu Ru Kou, PhD

Academic Editor

PLOS ONE

Journal Requirements:

“I have read the journal's policy and the authors of this manuscript have the following competing interests: [HE is affiliated with the Department of Healthcare Quality Assessment at the University of Tokyo. The department is a social collaboration department supported by grants from the National Clinical Database, Johnson & Johnson K.K., and Nipro Corporation.]”

Reviewers' comments:

Reviewer's Responses to Questions

**Comments to the Author**

1. Is the manuscript technically sound, and do the data support the conclusions?

Reviewer #1: Yes

Reviewer #2: No

2. Has the statistical analysis been performed appropriately and rigorously? 

Reviewer #1: Yes

Reviewer #2: Yes

3. Have the authors made all data underlying the findings in their manuscript fully available?

Reviewer #1: Yes

Reviewer #2: Yes

4. Is the manuscript presented in an intelligible fashion and written in standard English?

Reviewer #1: Yes

Reviewer #2: Yes

5. Review Comments to the Author

Reviewer #1: Thank you for giving me good opportunity to read Dr. Takekawa’s excellent paper entitled “Predict model for prolonged ICU stay using APACHE II, APACHE III and SAPS II scores: A Japanese multicenter retrospective cohort study. ”. This author clearly showed the risk factors for prolonged ICU stay in Japan based on JIPAD database, and this result would be very important to consider the contents of intensive care for risk patients, and also re-set up the health insurance service. So I think this paper would be appropriate enough for the publish in the PLOS ONE journal, however, to improve its perfection as a scientific paper, I request the author to add minor revision as following.

I’m looking forward to reading revised version shortly.

Again, thank you for giving me such a nice opportunity to review such an excellent paper.”

Major

1．Table 1 looks so long and contained too many information. I recommend to divide the table 1 appropriately like backgrounds including number, male, age, BMI, kinds of admission, outcome as 1 table, and diagnosis and chronic disease as 1 table, interventions as 1 table, and independent 1 table for about APACHE2, 3, and SAPSII.

2．The author succussed to approve models for predicting prolonged stays in Japanese ICU with appropriate statistical analysis, however there are too many long tables to understand not easy. I recommend two figures to show AUC for predicting prolonged ICU stay and hospital death with each model, if the author will accept and consider looking better. It’s up to the author.

Minor

1．Please add more detail explanation about JIPAD like as the number of participated facilities, on line-based input system, and past achievements, in page 4, lines 25 to Page4, line 2.

2．Please add full spell of MV as “mechanical ventilation” in the Page5, line 6.

3．Please re-check the expression of page in each reference of NO. 6 and 18.

Reviewer #2: This article entitled　“Predict models for prolonged ICU stay using APACHE II, APACHE III

and SAPS II scores: A Japanese multicenter retrospective cohort study.” reported by Daiki Takekawa et al. Author analyzed predict model for prolong ICU stay. I agree this retrospective multicenter cohort study showed some new factors regarding to prolonged ICU stay.

However, there are so many significant factors to predict the prolonged ICU stay.

So, we can not recognize the specific factor to predict the prolonged ICU stay.

Major comments

1.First, author should mention about the definition of prolonged ICU stay.

Why did you chose 14 days ? Author must show the definition and its reason,

not only referring just citations.

2. It is too difficult for clinician to understand usefuleness for this predictive models.

After all, which combination of factors is the best for predicting?

3.Author should use more figures for understanding. I think that ROC analysis is suitable for

this study. Please show visually ROC analysis by figure.

6. PLOS authors have the option to publish the peer review history of their article (what does this mean?). If published, this will include your full peer review and any attached files.

Reviewer #1: No

Reviewer #2: No

---

## [Author Response · Author response to Decision Letter 0]

14 Apr 2022

April 4, 2022

Submission no: PONE-D-22-01670

Submission title: Predict models for prolonged ICU stay using APACHE II, APACHE III and SAPS II scores: A Japanese multicenter retrospective cohort study

Dear Academic Editor Yu Ru Kou, PhD

Thank you for your letter of March 30 regarding the above manuscript. We are pleased to know that our manuscript will be reconsidered after revision according to your reviewers’ comments. Our responses to the reviewers’ comments are noted below.

Reviewer #1: 

Major 

Q1. Table 1 looks so long and contained too many information. I recommend to divide the table 1 appropriately like backgrounds including number, male, age, BMI, kinds of admission, outcome as 1 table, and diagnosis and chronic disease as 1 table, interventions as 1 table, and independent 1 table for about APACHE2, 3, and SAPSII. 

R1. As suggested, we divided the Table 1 to Table 1–4.

Q2．The author succussed to approve models for predicting prolonged stays in Japanese ICU with appropriate statistical analysis, however there are too many long tables to understand not easy. I recommend two figures to show AUC for predicting prolonged ICU stay and hospital death with each model, if the author will accept and consider looking better. It’s up to the author. 

R2. As suggested, we added Fig 3, 4 to show ROC curves for predicting prolonged ICU stay and hospital death with each model.

Minor 

Q1．Please add more detail explanation about JIPAD like as the number of participated facilities, on line-based input system, and past achievements, in page 4, lines 25 to Page4, line 2. 

R1. As suggested, we added more detail explanation about JIPAD in the materials and methods section (page 4, line 15–21). Detail explanation about JIPAD is below.

JIPAD was established by the Japanese Society of Intensive Care Medicine in 2014. 94 ICUs participate in JIPAD as of December 31, 2021. Each participating ICU submits data of their ICU patients, such as characteristics, severity scores at ICU admission, therapies in the ICU and outcomes, using on-line data system. Submitted data are routinely monitored and corrected by members of the JIPAD working group to improve the credibility of data. A recent study developed a mortality prediction model for adult patients admitted to ICU in Japan using JIPAD [13]. Further information about JIPAD can be found elsewhere [12].

12. Irie H, Okamoto H, Uchino S, Endo H, Uchida M, Kawasaki T, et al.; JIPAD Working Group in the Japanese Society of Intensive Care Medicine. The Japanese Intensive care PAtient Database (JIPAD): A national intensive care unit registry in Japan. J Crit Care. 2020; 55: 86-94. doi: 10.1016/j.jcrc.2019.09.004.

13. Endo H, Uchino S, Hashimoto S, Aoki Y, Hashiba E, Hatakeyama J, Hayakawa K, Ichihara N, Irie H, Kawasaki T, Kumasawa J, Kurosawa H, Nakamura T, Ohbe H, Okamoto H, Shigemitsu H, Tagami T, Takaki S, Takimoto K, Uchida M, Miyata H. Development and validation of the predictive risk of death model for adult patients admitted to intensive care units in Japan: an approach to improve the accuracy of healthcare quality measures. J Intensive Care. 2021; 9(1): 18. doi: 10.1186/s40560-021-00533-z.

Q2. Please add full spell of MV as “mechanical ventilation” in the Page5, line 6. 

R2. We are sorry for this mistake. We added it.

Q3. Please re-check the expression of page in each reference of NO. 6 and 18. 

R3. We are sorry for this mistake. We corrected it.

Reviewer #2:

Major comments 

Q1. First, author should mention about the definition of prolonged ICU stay. Why did you chose 14 days ? Author must show the definition and its reason, not only referring just citations. 

R1. As suggested, we added the reason why we chose this definition in the materials and methods section (page 4, line 26 to page 5, line 3). The reason is below.

It is because special fees for critical care are limited to a period of 14 days in the Japanese health insurance system, even if the status of the patient remains serious for more than 14 days. Additionally, previous studies about prolonged ICU stay that conducted in other countries also used this definition [2,14,15]. 

2. Arabi Y, Venkatesh S, Haddad S, Al Shimemeri A, Al Malik S. A prospective study of prolonged stay in the intensive care unit: predictors and impact on resource utilization. Int J Qual Health Care. 2002; 14(5): 403-10. doi: 10.1093/intqhc/14.5.403.

14. Laupland KB, Kirkpatrick AW, Kortbeek JB, Zuege DJ. Long-term mortality outcome associated with prolonged admission to the ICU. Chest. 2006; 129(4): 954-9. doi: 10.1378/chest.129.4.954.

15. Zampieri FG, Ladeira JP, Park M, Haib D, Pastore CL, Santoro CM, et al. Admission factors associated with prolonged (>14 days) intensive care unit stay. J Crit Care. 2014; 29(1): 60-5. doi: 10.1016/j.jcrc.2013.09.030. 

Q2. It is too difficult for clinician to understand usefuleness for this predictive models. After all, which combination of factors is the best for predicting? 

R2. Thank you for your variable comment. As you pointed out, there are a lot of variables in our models. Thus, we performed backward deletion to reduce variables in a stepwise manner. The results are shown in below table (Model 1–3). However, only “Male” was removed from each model, so our models still contained seven variables. Generally, predict models are not simple. Indeed, although previous studies also developed predict models for prolonged ICU stay, their models contain a lot of variables as well as the present study (2, 16). It is because these studies didn't intend to develop simple predict models but intended to develop accurate predict models as well as our study. Indeed, the AUC values for all our predict models were more than 0.8, which means that the discrimination abilities of these models were good. So, we did not change the table 5. The present models look complicated due to many categories, particularly in diagnosis, but there are only eight variables in these models. It is not so complicated.

Model 1

 aOR 95% CI P value

(Intercept) 0.007 0.006, 0.010 <0.001*

APACHE II 

–10 reference 

11–20 3.650 2.910, 4.590 <0.001*

21–30 12.60 9.940, 15.90 <0.001*

31–40 17.90 13.80, 23.00 <0.001*

41+ 9.050 6.500, 12.60 <0.001*

BMI (kg/m2) 

18.5 ≤, < 25 reference 

 < 18.5 0.953 0.848, 1.070 0.419

25 ≤, < 35 1.140 1.030, 1.270 0.013*

35 ≤ 1.330 0.936, 1.890 0.111

RRT/MET 1.240 1.060, 1.470 <0.001*

Postresuscitation 1.330 1.090, 1.610 <0.001*

Readmisson 1.290 1.100, 1.500 <0.001*

LOS before ICU admission (days) 

0 reference 

1–7 0.757 0.674, 0.850 <0.001*

8–14 0.970 0.821, 1.150 0.724

15+ 1.280 1.120, 1.470 <0.001*

Diagnosis 

-Post-operative 

Cardiovascular disease reference 

Respiratory disease 0.209 0.146, 0.301 <0.001*

Digestive disease 0.550 0.466, 0.650 <0.001*

Neurological disease 0.675 0.556, 0.820 <0.001*

Trauma 1.260 0.879, 1.800 0.208

Metabolic/hematological disease 0.425 0.158, 1.140 0.091

Urinary disease 0.145 0.071, 0.293 <0.001*

Muscle/bone/skin disease 0.282 0.185, 0.430 <0.001*

Obstetrics/gynecological disease 0.177 0.073, 0.429 <0.001*

-Non-operative 

Cardiovascular disease 0.894 0.755, 1.060 0.200

Respiratory disease 2.080 1.790, 2.410 <0.001*

Digestive disease 1.170 0.950, 1.440 0.140

Neurological disease 1.280 1.040, 1.570 0.022*

Sepsis 1.020 0.807, 1.280 0.882

Trauma 1.490 1.080, 2.050 0.014*

Metabolic disease 0.333 0.214, 0.518 <0.001*

Hematological disease 1.133 0.698, 1.820 0.622

Urinary disease 0.779 0.456, 1.330 0.359

Muscle/bone/skin disease 2.030 1.380, 2.980 <0.001*

Others 1.260 0.790, 2.010 0.331

Model 2

 aOR 95% CI P value

(Intercept) 0.005 0.003, 0.006 <0.001*

APACHE III score 

–40 reference 

41–80 6.270 4.820, 8.160 <0.001*

81–120 21.20 16.20, 27.80 <0.001*

121–160 28.50 21.30, 38.20 <0.001*

161+ 12.50 8.530, 18.30 <0.001*

BMI (kg/m2) 

18.5≤, < 25 reference 

<18.5 0.952 0.847, 1.070 0.410

25 ≤, < 35 1.160 1.040, 1.290 0.006*

35 ≤ 1.370 0.965, 1.950 0.078

RRT/MET 1.250 1.060, 1.480 0.007*

Postresuscitation 1.250 1.020, 1.520 0.030*

Readmisson 1.240 1.060, 1.440 0.007*

LOS before ICU admission (days) 

0 reference 

1–7 0.774 0.689, 0.869 <0.001*

8–14 0.955 0.808, 1.130 0.593

15+ 1.290 1.130, 1.480 <0.001*

Diagnosis 

-Post-operative 

Cardiovascular disease reference 

Respiratory disease 0.231 0.161, 0.332 <0.001*

Digestive disease 0.513 0.434, 0.607 <0.001*

Neurological disease 0.846 0.695, 1.030 0.095

Trauma 1.470 1.020, 2.100 0.004*

Metabolic/hematological disease 0.511 0.189, 1.380 0.185

Urinary disease 0.156 0.078, 0.316 <0.001*

Muscle/bone/skin disease 0.285 0.187, 0.435 <0.001*

Obstetrics/gynecological disease 0.193 0.080, 0.468 <0.001*

-Non-operative 

Cardiovascular disease 0.969 0.817, 1.150 0.717

Respiratory disease 2.270 1.950, 2.630 <0.001*

Digestive disease 1.200 0.972, 1.480 0.091

Neurological disease 1.430 1.160, 1.760 <0.001*

Sepsis 1.050 0.835, 1.330 0.663

Trauma 1.710 1.250, 2.360 <0.001*

Metabolic disease 0.360 0.232, 0.560 <0.001*

Hematological disease 1.210 0.749, 1.960 0.434

Urinary disease 0.817 0.478, 1.400 0.459

Muscle/bone/skin disease 2.080 1.410, 3.060 <0.001*

Others 1.450 0.906, 2.310 0.122

Model 3

 aOR 95% CI P value

(Intercept) 0.004 0.003, 0.005 <0.001*

SAPS II score 

–25 reference 

26–50 7.590 6.200, 9.300 <0.001*

51–75 22.60 18.30, 28.00 <0.001*

76–100 21.30 16.60, 27.30 <0.001*

101+ 6.830 4.090, 11.40 <0.001*

BMI (kg/m2) 

18.5 ≤, < 25 reference 

< 18.5 0.963 0.856, 1.080 0.523

25 ≤, < 35 1.150 1,030, 1.270 0.001*

35 ≤ 1.370 0.965, 1.960 0.078

RRT/MET 1.220 1.030, 1.430 0.019*

Postresuscitation 1.360 1.120, 1.660 0.002*

Readmisson 1.200 1.030, 1.400 0.020*

LOS before ICU admission (days) 

0 reference 

1–7 0.905 0.807, 1.010 0.086

8–14 1.120 0.945, 1.320 0.194

15+ 1.520 1.320, 1.740 <0.001*

Diagnosis 

-Post-operative 

Cardiovascular disease reference 

Respiratory disease 0.298 0.207, 0.429 <0.001*

Digestive disease 0.575 0.486, 0.680 <0.001*

Neurological disease 0.862 0.707, 1.050 0.141

Trauma 1.270 0.885, 1.810 0.197

Metabolic/hematological disease 0.591 0.218, 1.600 0.300

Urinary disease 0.223 0.110, 0.450 <0.001*

Muscle/bone/skin disease 0.351 0.230, 0.537 <0.001*

Obstetrics/gynecological disease 0.267 0.110, 0.647 <0.001*

-Non-operative 

Cardiovascular disease 0.915 0.774, 1.080 0.301

Respiratory disease 2.330 2.010, 2.700 <0.001*

Digestive disease 1.300 1.050, 1.590 0.014*

Neurological disease 1.340 1.090, 1.650 0.006*

Sepsis 1.150 0.913, 1.450 0.236

Trauma 1.620 1.180, 2.230 0.003*

Metabolic disease 0.384 0.247, 0.597 <0.001*

Hematological disease 1.300 0.802, 2.100 0.289

Urinary disease 0.946 0.554, 1.620 0.840

Muscle/bone/skin disease 2.400 1.630, 3.540 <0.001*

Others 1.540 0.966, 2.450 0.069

2. Arabi Y, Venkatesh S, Haddad S, Al Shimemeri A, Al Malik S. A prospective study of prolonged stay in the intensive care unit: predictors and impact on resource utilization. Int J Qual Health Care. 2002; 14(5): 403-10. doi: 10.1093/intqhc/14.5.403.

16. Kramer AA, Zimmerman JE. A predictive model for the early identification of patients at risk for a prolonged intensive care unit length of stay. BMC Med Inform Decis Mak. 2010; 10: 27. doi: 10.1186/1472-6947-10-27.

Q3. Author should use more figures for understanding. I think that ROC analysis is suitable for this study. Please show visually ROC analysis by figure. 

R3. As suggested (reviewer 1 also suggested), we added Fig 3, 4 to show ROC curves for predicting prolonged ICU stay and hospital death with each model.

We thank the reviewers for their comments and hope that this paper is now suitable for publication in PLOS ONE.

Sincerely yours,

Daiki Takekawa, M.D, PhD.

---

## [Decision Letter · Decision Letter 1]

27 May 2022

Predict models for prolonged ICU stay using APACHE II, APACHE III

and SAPS II scores: A Japanese multicenter retrospective cohort study

PONE-D-22-01670R1

Dear Dr. Takekawa,

We’re pleased to inform you that your manuscript has been judged scientifically suitable for publication and will be formally accepted for publication once it meets all outstanding technical requirements.

Kind regards,

Yu Ru Kou, PhD

Academic Editor

PLOS ONE

Additional Editor Comments (optional):

Reviewers' comments:

Reviewer's Responses to Questions

**Comments to the Author**

1. If the authors have adequately addressed your comments raised in a previous round of review and you feel that this manuscript is now acceptable for publication, you may indicate that here to bypass the “Comments to the Author” section, enter your conflict of interest statement in the “Confidential to Editor” section, and submit your "Accept" recommendation.

Reviewer #1: All comments have been addressed

2. Is the manuscript technically sound, and do the data support the conclusions?

Reviewer #1: Yes

3. Has the statistical analysis been performed appropriately and rigorously? 

Reviewer #1: Yes

4. Have the authors made all data underlying the findings in their manuscript fully available?

Reviewer #1: Yes

5. Is the manuscript presented in an intelligible fashion and written in standard English?

Reviewer #1: Yes

6. Review Comments to the Author

Reviewer #1: Thank you for the prompt reply for revising your excellent paper. The second version was fully revised appropriately, especially, additional figure and revised tables looks excellent, so I think now this paper would be good enough to be published in PLOS ONE journal. Finally, thank you for giving me good opportunity to review your paper.

7. PLOS authors have the option to publish the peer review history of their article (what does this mean?). If published, this will include your full peer review and any attached files.

Reviewer #1: No

---

## [Editor Report · Acceptance letter]

31 May 2022

PONE-D-22-01670R1 

Predict models for prolonged ICU stay using APACHE II, APACHE III and SAPS II scores: A Japanese multicenter retrospective cohort study 

Dear Dr. Takekawa:

I'm pleased to inform you that your manuscript has been deemed suitable for publication in PLOS ONE. Congratulations! Your manuscript is now with our production department. 

Kind regards, 

on behalf of

Dr. Yu Ru Kou 

Academic Editor

PLOS ONE